# The *MyoD1* Promoted Muscle Differentiation and Generation by Activating *CCND2* in Guanling Cattle

**DOI:** 10.3390/ani12192571

**Published:** 2022-09-26

**Authors:** Di Zhou, Yan Wang, Rong Yang, Fu Wang, Zhonghai Zhao, Xin Wang, Lingling Xie, Xingzhou Tian, Guoze Wang, Bo Li, Yu Gong

**Affiliations:** 1Guizhou Testing Center for Livestock and Poultry Germplasm, Guiyang 550018, China; 2Key Laboratory of Animal Genetics, Breeding and Reproduction in the Plateau Mountainous Region, Ministry of Education, Guizhou University, Guiyang 550025, China; 3Guizhou Livestock and Poultry Genetic Resources Management Station, Guiyang 550001, China

**Keywords:** Guanling cattle, CRISPR/CAS9, transcriptome, *MyoD1*

## Abstract

**Simple Summary:**

Muscle proliferation and differentiation is a complex process, which is mainly regulated by the myogenic regulatory factors (*MRFs*) gene family. As one of its family members, myogenic differentiation 1 (*MyoD1*) is an important regulator of myoblast differentiation and fibrogenesis, and its main function is to regulate the proliferation of myoblasts and the directed development of myogenic cell lines. In this study, CRISPR/CAS9 technology was used to construct the *MyoD1* gene knockout MDBK cell lines, and then the transcriptome of MDBK cells was analyzed based on high-throughput sequencing technology. The differentially expressed genes in the transcriptome and the functional enrichment of differentially expressed genes were analyzed. The differentially expressed genes in the transcriptome and the functional enrichment were analyzed to provide experimental data for further studies on the mechanism of *MyoD1* regulating *CCND2* in myocyte differentiation.

**Abstract:**

The purpose of this study was to analyze the transcriptome of *MyoD1* gene knockout MDBK cells (bovine kidney cells) using high-throughput sequencing. For the first time, CRISPR/CAS9 technology was used to construct a *MyoD1* knockout in MDBK cells and transcriptome sequence analysis was used to examine *MyoD1*-related target gene expression. Transcriptome sequencing indicated the presence of 723 differentially expressed genes (DEGs) by comparing wild type and *MyoD1* knockout MDBK cells and included 178 upregulated and 72 downregulated genes. The DEGs are mainly enriched in Pl-3-kinase and AKT, p53 signaling pathways. Quantitative RT-PCR confirmed that *PDE1B*, *ADAMTS1*, *DPT,* and *CCND2* were highly expressed in the leg muscle, longissimus dorsi, and shoulder of Guanling cattle, and *CCND2* was inhibited after *MyoD1* knockout, suggesting it may be a key downstream gene of *MyoD1* and associated with muscle formation and differentiation in Guanling cattle. This provides experimental data for subsequent studies on the regulatory mechanisms of muscle differentiation in Guanling cattle.

## 1. Introduction

Muscle development is regulated by members of the myogenic regulatory factor (MRF) gene family that regulate the proliferation and differentiation of muscle cells and fibers during embryonic as well as postnatal development. Myogenic differentiation 1 (*MyoD1*), an MRF member, is a key regulator of myoblast differentiation and muscle fiber formation [1]. *MyoD1* is a basic helix-loop-helix (bHLH) protein that acts in the nucleus and possesses an amino-terminal transcriptional activation domain and a carboxyl-terminal α-helical domain [2,3]. The protein is activated by Ser, Thr, and Tyr phosphorylation. The bHLH domain specifically binds and interacts with proteins possessing helix-loop-helix domains, and the adjacent basic region is necessary to bind to the promoters or enhancers of muscle-specific genes such as *CK* and *myogenin*. The presence of *MyoD1* can promote the specific transcription of target genes by forming an active heterodimer with proteins of the E protein family containing the bHLH domain and then combining with the conserved sequence E-box of the promoter or enhancer region of the target gene [4,5].

*MyoD1* functions as the primary switch for muscle-specific gene activation and transcription and can recruit the histone lysine methyl- and acetyltransferases Set7 and p300 that perform H3K4me1 and H3K27ac modifications necessary for stem cell differentiation into myoblasts. In addition, *MyoD1* also regulates muscle development through cooperation with the transcription factors c-Jun, Jdp2, Meis, and Runx1 that then enhance muscle-related gene expression [6]. For example, when GFP was fused with the bFGF gene, it was found that GFP positive myoblasts existed in the recipient muscle, which promoted the ability of muscle fiber regeneration and significantly increased the expression of myofibril protein [7]. *MyoD1* also induces expression of miR-494-3p that inhibits p300 gene expression and myosin (MYH2) expression in multifunctional stem cells during skeletal muscle development and inhibits skeletal muscle differentiation [8]. Currently, significant progress has been made in the study of the *MyoD1* gene in pigs, mice, and chickens [9,10,11], but it is difficult to figure out how to promote bovine muscle cell production.

Guanling cattle are a local fine breed in Guizhou Province, one of the 78 local livestock breeds under national key protection, ranking first among the four major yellow cattle in Guizhou with a high reproduction rate, high slaughter rate, high meat yield, high amino acid content, high protein content, low fat content, and other characteristics, with high food value, economic value, and development value. Previous studies of Guanling cattle have utilized promoter and transcription factor analyses combined with yeast one-hybrid systems to demonstrate that Myf6 strongly activated the *MyoD1* promoter [12]. However, the role of *MyoD1* in muscle differentiation is unclear. In our pre-experiments, we found that knockdown of the *MyoD1* gene inhibited the growth of muscle cells in Guanyang cattle, so we developed an experimental system employing bovine kidney cells (MDBK) to examine the effects of a *MyoD1* knockout more closely. We employed the CRISPR/CAS9 technique and used transcriptome analysis of *MyoD1* knockout MDBK cells to identify differentially expressed genes (DEG). Our research focus was to understand the role *MyoD1* plays in muscle cell differentiation.

## 2. Materials and Methods

### 2.1. Cultivation of MDBK

MDBK was purchased from the Shanghai Institute of Cell Biology, Chinese Academy of Sciences, and was cultured in high-sugar DMEM medium.

### 2.2. sgRNA Construction

The bovine *MyoD1* gene sequence (Acc. No. NM_001040478.2) was used to design CRISPR gene knockouts using the online resource http://www.e-crisp.org/ (accessed on 10 April 2021). The 4 sgRNAs possessing the highest specificity and annotation scores were synthesized by Jin Kairui Bioengineering (Wuhan, China) (see Table 1). Primers for *MyoD1* sgRNA were combined with DNA oligonucleotide annealing buffer and the LentiCrispr v2 vector (YaJi Biological of Shanghai, China) at 16 °C for 12 h using directions provided by the kit manufacturer (Thermo Fisher Scientific, Waltham, MA, USA). The linked products were transformed into *Escherichia coli* DH5α competent cells (Sangon Biotech, Shanghai, China) and single colonies were picked from LB ampicillin plates. Recombinant plasmids were extracted using an endotoxin-free extraction kit (Omega Bio-Tek, Dallas, TX, USA) and commercially sequenced to confirm their identity (Wuhan Bio-Engineering).

### 2.3. MyoD1 Gene Knockouts and Validation

We utilized lentivirus packaging of the plasmid constructs since preliminary experiments with MDBK cells (Institute of Cell Biology, Chinese Academy of Sciences, Shanghai, China) [13] had indicated that Lipofectamine 2000 (Invitrogen) transfection efficiencies were <5%. We therefore co-transfected 293T cells (Shanghai Biological Technology Co., Ltd. Enzymere, Shanghai, China) with the lentivirus packaging plasmid psPAX2 and lentiviral vector pMD2.G at 2:1:1) that were then incubated for 5 h according to the manufacturer’s specifications (Omega Bio-Tek, Dallas, Texas, USA). The medium was exchanged, and incubation was continued for 48 h. The lysed cells were filtered (0.45 μm) to obtain a 6 mL virus solution, and 3 mL was added to MDBK cells in a 6-well plate (4 sg types in total) and incubated for 8 h. The medium was changed, and the cells were incubated for an additional 24 h. The cells were then re-infected with the remaining 3 mL of virus solution and incubated as per above. The cells were added to fresh medium containing 2 μg/mL puromycin (ChemeGen, Los Angeles, CA, USA) and incubated for 60 h. Samples were taken for genomic DNA extraction and the remainder of the cells were expanded to 24-well plates and genomic DNA was again extracted to identify the knockout effect using PCR. PCR reactions utilized 2 × Taq PCR Master Mix (Sangon Biotech, Shanghai, China) using 10 pmol of primer and 0.5 μL of template DNA. The amplification conditions were as follows: 95 °C for 5 min, followed by 30 cycles of 95 °C for 30 s, 58 °C for 30 s, and 72 °C for 20 s, with a final incubation at 72 °C for 5 min. The amplicons were visualized by electrophoresis in 2% agarose gels and stained with SYBR green. Further verification using qRT-PCR utilized a commercial SYBR green amplification kit (Bio-Rad Laboratories, Hercules, CA, USA) containing 5 pmol of each primer (Table 2) and 1 μL template DNA. The reactions were carried out at 95 °C for 1 min, then 40 cycles at 95 °C for 15 s, and annealing at 55 °C for 30 s.

### 2.4. cDNA Library Construction and Sequencing

Trizol was used for RNA extraction using the protocol of the manufacturer (Invitrogen, Carlsbad, CA, USA). Total RNA was extracted from wild type and *MyoD1* knockout MDBK cells and mRNA was isolated using oligo dT magnetic beads (BioMag, Wuxi, China). The mRNA was fragmented into 200–300 nucleotide (nt) lengths using ion disruption (Shanghai Xinfan Technology Co., Ltd., Shanghai, China) and used as cDNA templates that were amplified using 6 nt random primers and reverse transcription (AXYGEN, Silicon Valley, USA). Second strand cDNA was synthesized following a similar procedure using the first-strand cDNA as template. These amplified fragments were used to construct the library that was sequenced by Nanjing Paterno Gene Technology (Nanjing, China) using the Illumina HiSeq platform (Illumina, San Diego, CA, USA) and tested for quality by an Aglent 2100 Bioanalyzer.

### 2.5. Enrichment of Differentially Expressed Genes

Clean reads were obtained by filtering the original sequencing data (Cutadapt) compared with the bovine genome (Bos_taurus.UMD3.1.dna.toplevel.fa) to obtain the gene sequence (Trimmomatic 0.39). Differentially expressed genes (DEG) were analyzed by the number of reads per 1000 bases of a gene per million reads (FPKM) and defined with DESeq software (false discovery rate (FDR) ≤ 0.01, absolute p value of Log2 ratio of ≥1). All genes were mapped to each term in the Gene Ontology (GO) database, and the number of differential genes in each term was calculated. Based on the whole genome, a hypergeometric distribution was used to calculate the terms and pathways of significantly enriched genes.

### 2.6. Validation of Differential Gene Expression

Heart, liver, spleen, lung, kidney, leg muscle, latissimus dorsi, and shoulder muscle tissues were collected from the Guanling cattle breeding farm in Guizhou province, and the amount of each tissue was about 0.25 g. The collected tissues were treated with PBS solution (10,000 U/mL penicillin and 10 mg/mL streptomycin) and put into 1.5 mL EP tubes, which were quickly transferred to the laboratory after being put into liquid nitrogen to extract tissue RNA using Trizol (Invitrogen, Carlsbad, CA, USA). After RT-PCR validation of the most significantly differential genes enriched, qPCR validation was then performed on the Guanling cattle tissue.

All animal experiments conformed to the Guizhou University Institutional Animal Care and Use Committee guidelines (Guizhou, China). The *β-actin* gene was used as a loading control reference gene (Table 3). All experiments were performed with 3 biological replicates and 3 technical replicates. The relative expression levels of the selected genes were calculated using the 2^−∆∆Ct^ method [14].

### 2.7. Statistical Analysis

The data were analyzed using SSPS 18.0 software (IBM, Chicago, Ill, USA). Significant differences were determined using a one-way ANOVA F-test at *p* < 0.05. DESeq2 was used to analyze the difference in gene expression and the condition of screening DEGs was multiple of differential expression where |log Fold change| > 1, at a significance of *p* < 0.05. Volcano plots for DEGs were drawn using the R language ggplot2 software package.

## 3. Results

### 3.1. MyoD1 Gene Knockouts

Our procedure for *MyoD1* gene knockouts utilized a plasmid vector encoding puromycin resistance. Therefore, we conducted preliminary experiments of the selection procedure and added puromycin to wild type MDBK cells to evaluate the lowest concentration that could completely kill the cells. We found that 2 μg/mL resulted in almost complete cell death by 48 h (Appendix A). We also examined whether Lipofectamine 2000 could be used for direct transfection of the CRISPR plasmids, and we utilized a commercial plasmid GFP vector to test this. However, the lipofectamine transfection efficiency of MDBK cells was <5%, so this direct avenue of gene introduction into these cells was abandoned (Appendix A).

We subsequently utilized the lentivirus infection method and infected MDBK cells with the lentivirus constructs for 48 h and then analyzed the cells for evidence of *MyoD1* gene interruption. We found that PCR amplification of genomic DNA samples from cells using four different targets generated ~700 bp amplicons that indicated an overall knockout effect (Appendix A). These amplicons were subsequently sequenced and the target for sgRNA6 generated the most unambiguous interruption of the *MyoD1* gene (Appendix A). We therefore expanded the gRNA6 cells and generated clones by plating at 1 cell per well in 96-well plates (limiting dilution analysis). We acquired a total of 20 clones after 2 weeks of culture. Target gene mutations were present in eight of the new cell lines, and we selected four obvious mutations to confirm the addition/deletion/mutation of each allele (Appendix A). The deletions in the *MyoD1* coding regions occurred in clones 3 and 22 (Figure 1). These results indicated that the *MyoD1* gene was successfully knocked out in these clones obtained from limiting dilution analysis. These results were corroborated by measurements of *MyoD1* steady state mRNA levels, and *MyoD1* was significantly downregulated in three mutants compared with wild type cells (Figure 2).

### 3.2. Transcriptome Sequencing

We utilized the clone gRNA6 cells for genomic analysis and procured >40 million clean reads for both wild type and *MyoD1* knockout MDBK cells, and high-quality read percentages were accessed by a comparison to the reference genome. The total number of sequences was >36 million and the alignment rate was 89% (Table 4). In addition, transcriptome analysis indicated >4000 new transcripts, 4600 generated by variable splicing and >36,000 SNPs in wild type MDBK cells and 6300 new transcripts, 4500 variable splicing, and 38,000 SNPs in *MyoD1* knockout MDBK cells (Table 5). By using clustering analysis of all samples (Figure 3), we can infer that *MyoD1* knockout was effectual, and some genes were used to perform difference and significance analysis (Pheatmap).

The data were utilized to generate DEG populations, and we identified 723 total DEGs. These included 370 upregulated and 353 downregulated genes in the experimental group compared with the control group (Figure 4).

### 3.3. GO Functional Annotation of DEGs

The DEGs were mapped using GO analysis by calculating the number of DEGs in each term. Employing the whole genome as the background, a hypergeometric distribution was used to calculate the term with significant enriched DEGs. The enrichment results indicated 668 DEGs that were involved in cellular components (CC), molecular function (MF), and biological processes (BP). The BP group consisted primarily of cell differentiation throughout biological processes; it was exclusively focused on muscle cell differentiation and muscle structure development (Figure 5 and Appendix A).

### 3.4. KEGG Enrichment Analysis of DEGs

All transcripts from wild type and mutant were also annotated to the KEGG database and the number of DEGs at different levels of the KEGG pathway were then counted to determine the primary metabolic and signaling pathways for these DEGs. The results indicated that 25,644 genes were enriched to the KEGG pathway, and this included 1417 DEGs. An analysis of the 20 most significant pathways revealed that these differential genes were enriched in Pl-3-kinase and AKT, p53 signaling pathways (Figure 6).

### 3.5. DEG Verification

Finally, based on the results of highly abundant and significant differences in cell differentiation, ten genes that showed the most significant differences in expression were selected to confirm whether the steady state transcriptome data correlated with quantitative RT-PCR measurements. The expressions of *PDE1B*, *Histone H3.1*, *ADAMTS1*, *DPT*, and *CYP4F2* were significantly upregulated (Figure 7a) and *WNT7A*, *SEMA3A*, *HOXD10*, *CCND2,* and *HR* were significantly downregulated in the *MyoD1* knockout group (Figure 7b). This is consistent with the cluster analysis results.

### 3.6. DEGs in Different Tissues of Guanling Cattle

In RT-PCR tissue validation, we found that *PDE1B, Histone H3.1, ADAMTS1, DPT, CYP4F2, WNT7A, SEMA3A, HOXD10, CCND2,* and *HR* genes were distributed in all tissues (Figure 8), but *PDE1B*, *ADAMTS1*, *DPT,* and *CCND2* were significantly more expressed in leg muscles, dorsalis longus, and shoulder compared to other genes (Figure 8a,c,d,j).

## 4. Discussion

Muscle growth and development are usually regulated by specific core genes and signal transduction pathways [15,16], and the *MyoD* gene family (*MyoD1, MYF5, MyoG,* and *MYF6*) are key regulators that control the expression of specific proteins in muscle cell proliferation and differentiation [17,18,19,20]. *MyoD1*, as a member of its family, is a major transcriptional regulator of muscle-specific genes, and its activity is directly related to muscle growth. Although previous studies related to the promoter binding site of the *MyoD1* gene in Guanling cattle have been reported [12,21,22], the mechanism of action of the bovine *MyoD1* gene on muscle cells has rarely been explored. In view of the cell growth arrest caused by *MyoD1* knockdown in muscle cells in previous experiments, in the present study, after knockdown of the *MyoD1* gene in bovine kidney cells, we discovered that *PDE1B, histone 3.1, ADAMTS1, DPT,* and *CYP4F2* were upregulated, implying they may be a compensatory pathway after *MyoD1* knockdown. In contrast, *WNT7A, SEMA3A, HOXD10, CCND2,* and *HR* were significantly downregulated, so we inferred that they might be downstream genes of the *MyoD1* gene.

To clarify the relationship between these genes and economic traits in Guanling cattle, we examined their expression in visceral tissues (heart, liver, spleen, lung, kidney) and skeletal muscles (leg muscles, dorsal longus, shoulder), respectively, and found that all 10 genes were expressed in the tissues, but *PDE1B, ADAMTS1, DPT,* and *CCND2* were expressed at significantly higher levels in leg muscles, dorsal longus, and shoulder than several other genes. This is also consistent with previous reports that *PDE1B* is associated with fat thickness and muscle in the latissimus dorsi region [23], *ADAMTS1* plays a regulatory role in muscle regeneration after injury [24], and *DPT* enhances cell adhesion, reduces cell proliferation, and promotes myogenic hyperplasia [25]. *PDE1B*, *ADAMTS1*, and *DPT* were elicited after knockdown of the *MyoD1* gene. The reason may be that they have a compensatory relationship in the regulation of muscle formation. In contrast, *CCND2* was significantly downregulated after *MyoD1* knockdown, with expression at the lowest level among the 10 genes but the highest expression in leg muscle tissue, suggesting that it is a downstream gene of the *MyoD1* gene that promotes muscle cell growth and differentiation. This is consistent with the report that *CCND2* is a novel key regulator of terminal differentiation of muscle progenitor cells [26], and we speculate that *MyoD1-CCND2* in promoting cell growth, proliferation, and metabolism through the Pl3K-AKt signaling pathway.

In summary, we suggest that *CCND2* is a key downstream gene of *MyoD1* and is closely associated with the involvement in muscle formation and differentiation in Guanling cattle through the Pl3K-AKt signaling pathway. This provides experimental data for subsequent studies on the regulatory mechanisms of muscle differentiation in Guanling cattle.

## 5. Conclusions

In this study, the MDBK cell line possessing a *MyoD1* gene knockout was success-fully constructed using the CRISPR/CAS9 technique, and differentially expressed genes were identified by transcriptome sequencing and screening. These DEGs were significantly enriched in GO terms and KEGG pathways like *PDE1B, ADAMTS1,* and *CCND2*, suggesting that *MyoD1* affects muscle cell differentiation and muscle growth mainly by regulating the action of these pathways. The PCR results also confirmed this inference.

## Figures and Tables

**Figure 1 animals-12-02571-f001:**
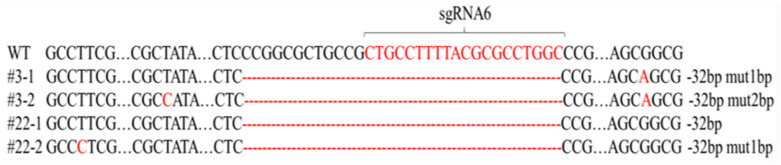
Sequencing results for 4 *MyoD1* gene disruptions using sgRNA6.

**Figure 2 animals-12-02571-f002:**
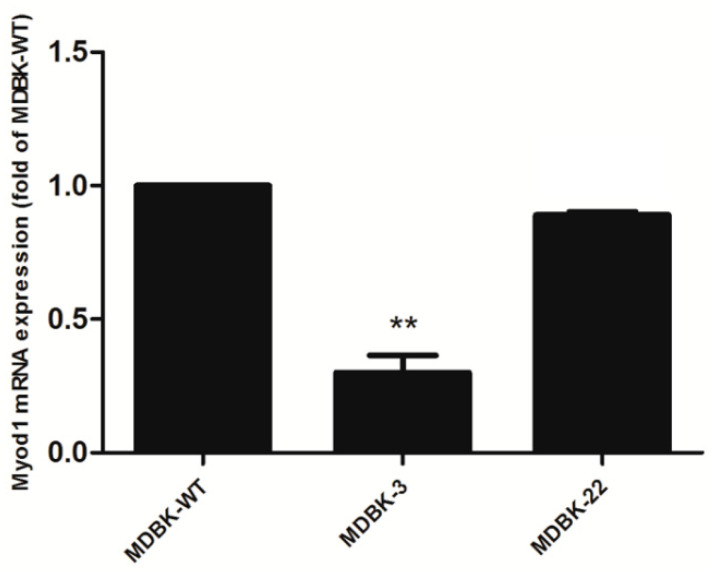
Real time qRT-PCR analysis of *MyoD1* gene expression for the two selected cloned cell lines. ** *p* < 0.01.

**Figure 3 animals-12-02571-f003:**
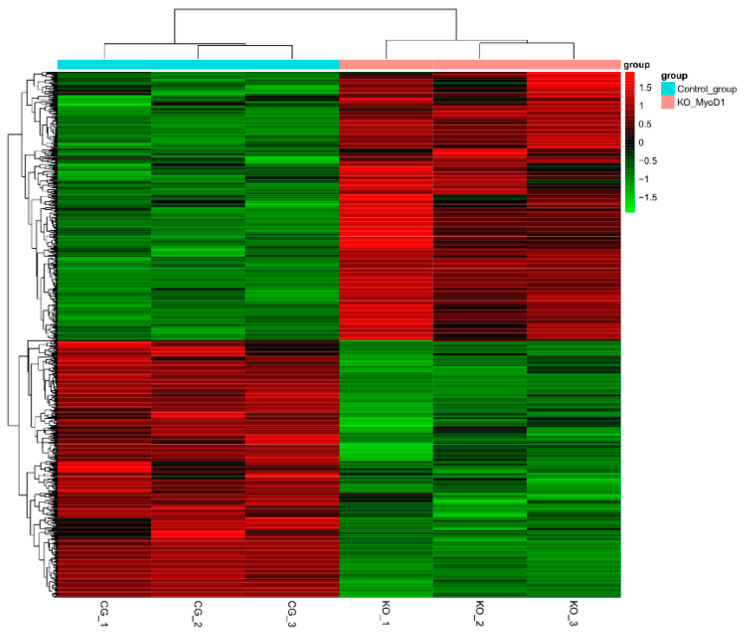
Clustering analysis of all samples.

**Figure 4 animals-12-02571-f004:**
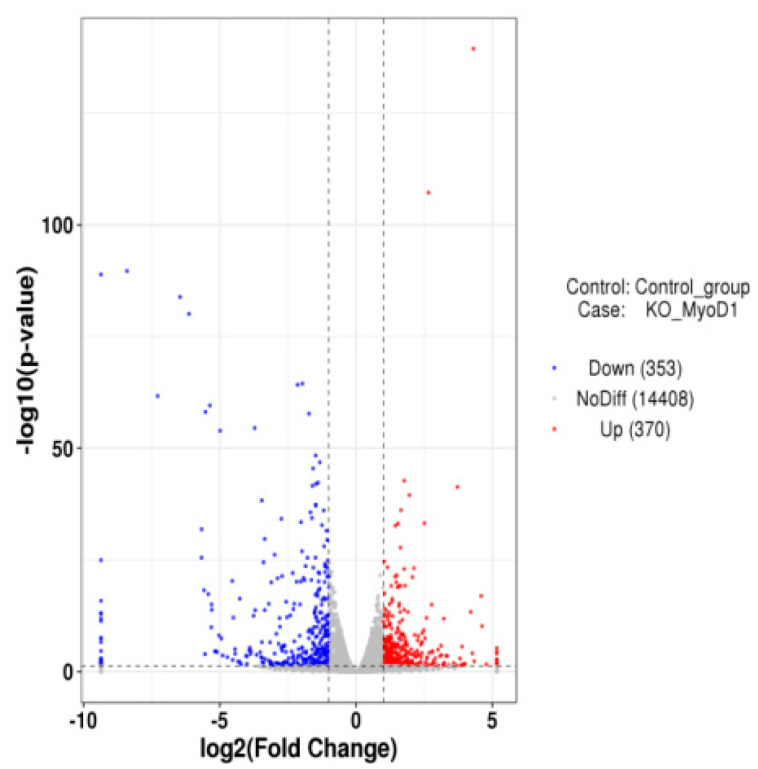
DEGs generated from a comparison of MDBK wild type and *MyoD1* knockout cells. Red, upregulated; blue, downregulated.

**Figure 5 animals-12-02571-f005:**
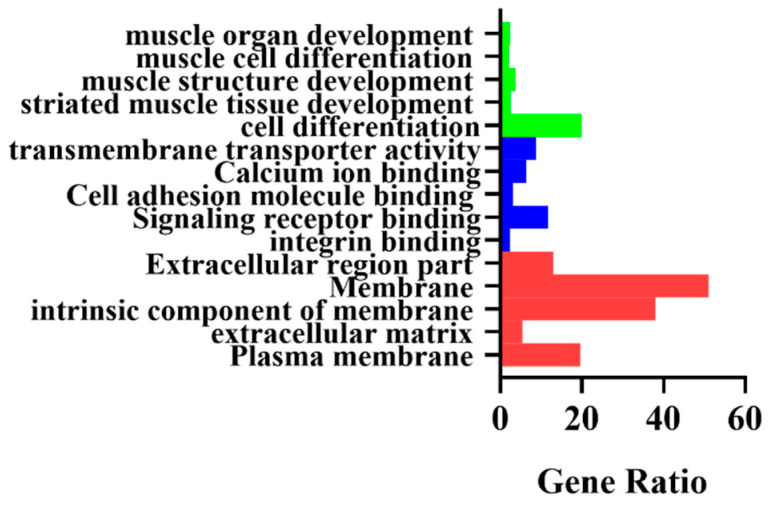
GO enrichment analysis of DEGs. Green bars are biological processes, blue bars are molecular functions, and red bars are cellular components.

**Figure 6 animals-12-02571-f006:**
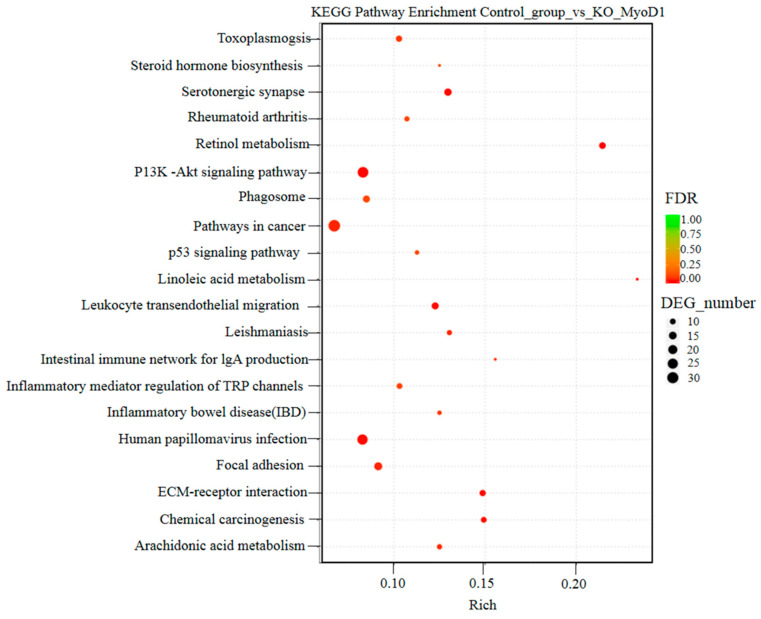
KEGG enrichment analysis of DEGs.

**Figure 7 animals-12-02571-f007:**
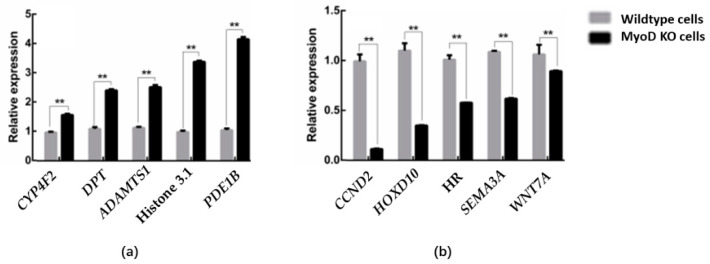
Verification of differentially expressed genes by qRT-PCR compared with beta-actin. (**a**) upregulation of gene expression levels. (**b**) downregulation of gene expression levels. ** *p* < 0.01.

**Figure 8 animals-12-02571-f008:**
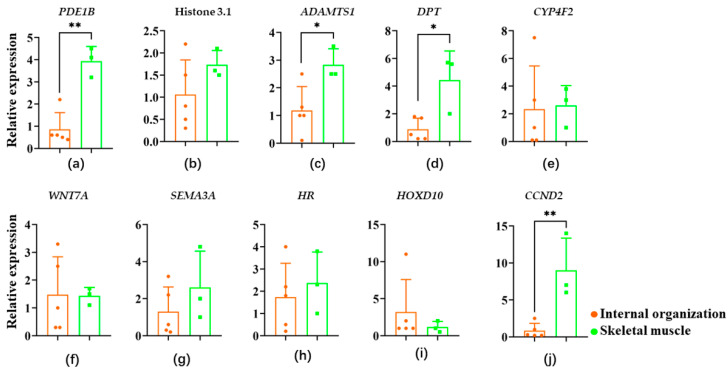
Quantification of mRNA expression levels in tissues of Guanling cattle. (**a**) *PDE1B*. (**b**) Histone 3.1. (**c**) *ADAMTS1*. (**d**) *DPT*. (**e**) *CYP4F2*. (**f**) *WNT7A*. (**g**) *SEMA3A*. (**h**) HR. (**i**) *HOXD10*. (**j**) *CCND2*. * *p* < 0.05, ** *p* < 0.01.

**Table 1 animals-12-02571-t001:** The primer sequence of *MyoD1* sgRNA.

Name	Primer	Sequence (5′→3′)
*MyoD1*-sgRNAg5	5For	CACCGCATGGTAGCAGCCTTGCGG
	5Rev	AAACCCGCAAGGCTGCTACCATGC
*MyoD1*-sgRNAg6	6For	CACCGCCAGGCGCGTAAAAGGCAG
	6Rev	AAACCTGCCTTTTACGCGCCTGGC
*MyoD1*-sgRNAg7	7For	CACCGACTAACGCCGACCGCCGCA
	7Rev	AAACTGCGGCGGTCGGCGTTAGTC
*MyoD1*-sgRNAg8	8For	CACCGCAGCGTTTGAGCGTCTCGA
	8Rev	AAACTCGAGACGCTCAAACGCTGC

**Table 2 animals-12-02571-t002:** *MyoD1* primers.

Name	Sequence (5′→3′)
*MyoD1*-PCR	For.ACAGCGGACGACTTCTATGATGACCC
	Rev.AGCTCCTTGCCCTCTCGTAAACACAT
*MyoD1*-qRT-PCR	For.CTTCTTCGAGGACCTGGATC
	Rev.CGTTAGTCGTCTTGCGTTTG
*GAPDH*	For.GTTCAACGGCACAGTCAAGGCA
	Rev.TCCACCACATACTCAGCACCAG

**Table 3 animals-12-02571-t003:** DEG primers.

Name	Accession Numbers	Temperature Melting ℃	Sequence (5′→3′)
*PDE1B*	NM_174415.2	59.97	For.GGTGCTTTGATGTCTTTT Rec.GTTGGTAGTGCCAGTGTG
*Histone H3.1*	XM_002688505.5	60.04	For.CCCTGAGAGAGATCCGCCGTTAC, Rec.GATGTCCTTGGGCATGATGGTGA
*ADAMTS1*	NM_001101080.1	59.97	For.CGAACAGGAACTGGAAGCCTAAGAA, Rec.CCACGGAGAACAAGGTCAGAAGGTA
*DPT*	NM_001045903.1	59.97	For.GTGACGATGGGTGGGTGAA, Rec.CGAAGTAGCGGCTCTGGAA
*CYP4F2*	XM_010806565.3	60.03	For.CCTAAAAACATTGAATGGGACG, Rec.TGCTGATGAGGCAGATAACACC
*WNT7A*	NM_001192788.1	60.02	For.AGTGGGGTGGCTGCTCTGCCGACATC, Rec.CATGAGAGTCCGGGCGTTCTGTTTGA
*SEMA3A*	XM_015468793.2	60.01	For.ATTGTCTGTCTTTTCTGGGGAG, Rec.AGGAAGGTATGGTAACTGGAGC
*HOXD10*	NM_001099105.1	60.00	For.TGGCAGAGGTCTCCGTGTCC, Rec.CCAGCGTTTGGTGCTTAGTG
*CCND2*	XM_024992177.1	60.03	For.AGACCATCCCGCTGACCGCTGAGAA, Rec.GGTGACAGCCGCCAGGTTCCATTTC
*HR*	NM_001102535	60.03	For.GTGCCAGTTCCCTGATGCTC, Rec.TCCTGTTGGTTTCCCCGTTG
*β-actin*	NM_173979.3	60.00	For.GTCCACCTTCCAGCAGAT Rec.GCTAACAGTCCGCCTAGAA,

*PDE1B* is phosphodiesterase 1B, *ADAMTS1* is a disintegrin and metalloproteinase with thrombospondin motifs 1, *DPT* is dermatopontin, *CYP4F2* is cytochrome P450 family 4 subfamily F member 2, *WNT7A* is Wnt family member 7A, *SEMA3A* is semaphorin 3A, *HOXD10* is homeobox D10, *CCND2* is cyclin D2, and *HR* is Hairless.

**Table 4 animals-12-02571-t004:** Sequence results for *MyoD1* gene knockouts.

Sample	Total Reads	High-Quality Reads	Mapping Rate
KO_1	41,947,236	41,726,484	37,604,520 (90.12%)
KO_2	40,589,490	40,407,780	36,558,103 (90.47%)
KO_3	42,524,806	42,288,340	37,975,054 (89.80%)
CG_1	41,933,976	41,730,096	37,595,702 (90.09%)
CG_2	40,595,566	40,186,420	35,544,666 (88.45%)
CG_3	41,687,492	41,277,510	36,684,371 (88.87%)

KO: knockout; CG: control; triplicates are represented.

**Table 5 animals-12-02571-t005:** Transcriptome analysis.

Sample	Number of Novel Transcripts	Number of Alternative Splicing	SNP Number
KO_1	40,788	48,305	42,327
KO_2	35,380	46,581	36,696
KO_3	35,675	46,872	39,528
CG_1	36,990	45,802	38,655
CG_2	37,717	46,371	39,001
CG_3	37,260	47,553	40,287

KO: knockout; CG: control; triplicates are represented.

## Data Availability

The data set supporting the conclusions of this article is available by email of corresponding author.

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
