# Peer review of "The MyoD1 Promoted Muscle Differentiation and Generation by Activating CCND2 in Guanling Cattle"

_animals, 2022, doi:10.3390/ani12192571_

Round 1
Reviewer 1 Report (New Reviewer)

Author Response
Please see the attachment.

Reviewer 2 Report (New Reviewer)
I reviewed the manuscript entitled "The MyoD1 promoted muscle differentiation and generation by activating CCND2 in Guanling cattle" and I found it relevant to the scientific literature.
The introduction is well described and introduces the reader to the topic undertaken by the researchers.
Material and methods are properly and comprehensively described, which is necessary for this type of article.
The results are presented in a logical way and allow the reader to follow the course of the analysis. The figures especially figure 3 could be of better quality or moved into supplementary materials, the tables are informative. All calculations were carried out with the use of appropriate programs and methods, and were thoroughly described and presented in the results.
In the discussion, the authors discussed the results of the research in relation to the literature and touched upon the most important problems arising from the research. The discussion is written in a way that is understandable to the reader.
Author Response
/
This manuscript is a resubmission of an earlier submission. The following is a list of the peer review reports and author responses from that submission.
Round 1
Reviewer 1 Report
Overall, the authors have presented the study very well. Although there is is room to significantly improve the RNA-Seq data presented it in a more meticulous. This is essential for this type of study because the main conclusions are drawn from the transcriptome data, whether DEGs or even OMICs data. The study is missing correlation analysis (i.e. principle component analysis, and dendogram heatmap); furthermore, and in addition to the volcano plot, it would be also a visual asset to add the overall heatmap of the DEGs to get insights of raped changes into DEGs. These would serve as a good platform to further hypothesize future questions in in relation to the project.
These two analysis would be essential to complement the study.
Author Response
Dear Editors and Reviewers:
Thank you for your letter and for the reviewers’ comments concerning our manuscript entitled “Transcriptome analysis of MyoD1 gene knockout in MDBK cell lines based on high-throughput sequencing” (ID: 1698225). Those comments are all valuable and very helpful for revising and improving our paper, as well as the important guiding significance to our researches. We have studied comments carefully and have made correction which we hope meet with approval. Revised portion are marked in red in the paper. The main corrections in the paper and the responds to the reviewer’s comments are as flowing:

Reviewer 2 Report
General impressions
The manuscript address an interesting topic such as the study of the major myogenic regulator MyoD1.
There are some there are some aspects that need a revision though:
- Why the experimental model is based on MDKB cells and not, for instance, in muscle satellite cells?
- MyoD expression in KO cells versus expression in WT cells as assessed by qRT-PCR is different in one of the two selected clones (MDBK-3) but the difference seems quite unlikely in the other clone (MDBK-22), in spite of the significance stated in the graph. Could the authors give more information about the data and analysis?
- The 5 upregulated and 5 downregulated genes selected for qRT-PCR assessment might not be the most relevant concerning muscle cell differentiation and MyoD1 regulatory functions, even they showed the highest DEG scores, and their relationship with meat quality is quite difficult to establish. That last part of the study might need to be re-oriented.
- The manuscript describes first the methodological approach to obtain MyoD1 KO cells, but some of the results of the setting up of the technique should be included in the supplementary materials in order to better focus the manuscript.
Regarding some formal aspects, the manuscript should be revised for grammar and typographical errors (spacing for instance) and for tables and graphs captions.
Specific comments
- The rationale, hypothesis and purpose of the study should be more clearly established, including the in vivo part of the study: in the abstract, the purpose is defined as “to analyze the transcriptome of MyoD1 knockout MDBK cells”, but is difficult to follow from this to meat quality in Guanling cattle.
- Avoid generalities such as “linked them to cellular component, molecular function and biological processes” (abstract) and be specific.
- The pertinence of the cell model should be stated.
- In material and methods, there is no information about the in vivo experiment (animals, sampling, tissues…), it should be included.
- Some of the information referring the construction of the knockout cells (for instance, Figures 1 and 2) would be better in included in the supplementary materials in order not to lose the focus of the manuscript.
- Annealing temperature of qRT-PCR is missing (line 123)
- In section 2.3: There is no information about the quality of RNA (RIN?) or cDNA libraries (concentration, fragment size?), nor about the software used for the analysis. Phrase in lines 143-145 should be rewrited (…”for each up-regulated and down-regulated”…)
- In section 3.6. Explain the criteria for gene selection.
Nevertheless, it might be more appropriate to perform a biological pathway analysis and to analyze the molecular functions and biochemical pathways in order to map biological functions and thus better identify genes related with the role of MyoD1 in muscle cell differentiation (genes network).
- Results in Figure 7 need more data and explanation as expression in MDBK-22 and in the wild-type seem very similar and the graph is not very clear.
- In section 3.7. it is not clearly stated how the analysis of gene expression was performed, is there a calibrator tissue? The descriptions of the results is not very clear and the “expression patterns” not that evident. This section needs to be clarified.
- What do you mean with “pedigree potential of skeletal muscle”? (line 289)
- Rewrite the discussion in a way that it is not a mere description of the genes assessed by qRT-PCR highlighting its possible implication in muscle differentiation or carcass traits in beef cattle.
- Rewrite the conclusion in a way that is coherent with the aim and the results of the study. Do no focus in the possible relation of these 10 genes to meat quality traits as the design of the experiment does not allow to establish that kind of relationships.
- Please specify clearly the tables and graphs captions so they describe thoroughly the information in tables and graphs, considering that the abbreviations should be described. For instance, in the graphs in figures 10 and 11 the names of the genes have been shortcut but there is any explanation.
- In table 7, differentiate clearly the three GO ontology groups.
- Please avoid using the expressions “see above” and “see below” when referring to the methodology as it is very confusing. Indicate clearly where abouts in the text is the method you refer.
- Please revise for correct spacing all through the text, there are many missing spaces, for instance:
line 45: mation[1].MyoD1
line 54: gene[4,5]
- Please revise for correct grammar all through the text and correct errors such as:
Line 20: it is written “further study” instead of “further studies”
Author Response

(The authors gave the same response as above.)

Round 2
Reviewer 2 Report
The manuscript address an interesting topic such as the study of the major myogenic regulator MyoD1, using for that a model of MDKB in which MyoD1 expressions was knocked down by means of the novel technique CRISPR-Cas9 and further analysis of differentially expressed genes (in vitro). Five up-regulated and five down-regulated genes were selected for qPCR confirmation of the transcriptome results and the expression of those same 10 genes was relatively quantified in different tissues of Guanling cattle (in vivo). Although it is an interesting methodological approach, it is difficult to follow the connexion between the in vitro and the in vivo assays. It would help if the rationale, hypothesis and purpose of the study were more clearly established, including the in vivo part of the study. On the other hand, neither of the two assays presented in the manuscript seem to be “finished” as the results are poorly analysed in terms of getting to some conclusions. Therefore, the discussion section, and the conclusions, still need to be re-written.
Some of the up-regulated genes in the MyoD1 KO cells were most abundantly expressed in muscle than in other tissues such as lung or liver while some of the down-regulated genes in the MyoD1 KO cells were up-regulated in the muscular tissues, could the authors discuss this type of results?
Ÿ When I said that some of the information referring the construction of the knockout cells (for instance, Figures 1 and 2) would be better in included in the supplementary materials in order not to lose the focus of the manuscript” I was not meaning ALL the information about the KO cells but some, such as that related to the preliminary experiments and the selection of the transfection methods (Figures S1 and S2). It would be also interesting as well to include in this part, the justification for the selection of the MDBK cells as model due to the fact that grow of MyoD knocked out muscles cells was inhibited, which seems quite interesting and maybe something to further explore.
Figures S4 and S5 are also probably better in the supplementary material but Fig S6 and FigS7 fit well in the main text.
Ÿ There are two figures 3 and two figures 4, please correct.
In the second figure 3, it would be better to indicate “Wildtype cells” and “MyoD KO cells” that “test group” and “control group”. It would be also most appropriate to represent first the data of the wildtype data followed by the KO cells, in the X axis, than the other way round. The caption of the figure should be corrected: a) “expression of up-regulated genes” instead of “up-regulation of gene expression levels”; same for b).
In the second figure 4, the acronym of the genes should be above the graphs and the full names in the caption.
Ÿ When I indicated that you should “differentiate clearly the three GO ontology groups” in former table 7, now missing, I was meaning that the format of the table, that had a line in the middle, did not allow to see what functions corresponded to each process. The graph you have included to substitute the table is appropriate but still the table should be kept, maybe in the supplementary material.
Ÿ Can you provide some information about the results inferred from Figure 1 and include it in line 170?
Ÿ Could be any other biological enriched pathway worth of mentioning after of KEGG analysis (Figure 4)?
Ÿ When I said that in material and methods sections there were no information about the in vivo experiment (animals, sampling, tissues…) and it should be included, I was referring to the last part of the study, section 3.5, where results of the DEGs in tissues of Guanling cattle are explained but there no information about what tissues were sampled, how, from how many animals, etc. That is the information that should be included in “Material and Methods”.
Ÿ In Line 216 delete the sentence “The reference gene was beta-actin”, it is not necessary, it is supposed that the qPCR analysis was performed as indicated in section 2. What I was meaning with “it is not clearly stated how the analysis of gene expression was performed, is there a calibrator tissue? The descriptions of the results is not very clear and the “expression patterns” not that evident. This section needs to be clarified” has to do with my previous comment and the lack of information about the “in vivo” study, that is, the study with tissues taken from animals and no for the cell culture part of the study. The section is still confusing and the sentence “The genes identified as up-regulated and down-regulated were distributed within these expression patterns“ difficult to understand, what do you mean by “were distributed within these expression patterns”?
Ÿ Some of the modifications I suggested are the same in the revised version, although you said you have made them:
- Annealing temperature of qRT-PCR is STILL missing (line 123)
- In section 2.3: There is STILL no information about the quality of RNA (RIN?) or cDNA libraries (concentration, fragment size?), nor about the software used for the analysis. This is different from the reference gene and from the minimum information for PCR
- I indicated to avoid using the expressions “see above” and “see below” when referring to the methodology as it is very confusing. Those indications are still in the manuscript. In section 2. Materials and Methods, I think you should delete lines from 75 to 86 and include that information in the corresponding section instead of (see above). For instance, the “see above” in line 126 should be replaced by info in lines 80-81 about the method used for RNA extraction. And so forth.
In the line 49 you can delete “(see above)” as it is unnecessary.
In the line 103 you should change “see bellow” by “Supplementary material”
Ÿ In line 239 you include reference 15 as a previous study of yours, which seems to be a mistake.
Ÿ Regarding some formal aspects, the manuscript STILL should be revised, in spite that the authors said it was revised:
- Tittles and captions of figures and tables are not adequate, I recommend to take any other paper already published in Animals as an example to re-write them.
- Please revise for correct spacing all through the text, there are still many missing and incorrect spaces.
- Please revise for grammar and proper construction of sentences. As an example, but there are quite a few through the manuscript:
Lines 19-20: The sentence “To provide experimental data for further study on the mechanism of MyoD1 in myocyte differentiation” has no meaning in its own. Did you mean instead “The differentially expressed genes in the transcriptome and the functional enrichment of differentially expressed genes were analysed to provide experimental data for further study on the mechanism of MyoD1 in myocyte differentiation”?
Lines 21-22: please change “was to analyze the transcriptome of MDBK cells (bovine kidney cells) that possessed a MyoD1 gene knockout using high-throughput sequencing” by “was to analyze the transcriptome of MyoD1 gene knockout MDBK cells (bovine kidney cells) using high-throughput sequencing”
Line 34: please change “toward” by “on”
……………………….
- Please revise for typos and correct errors such as (but there are many more):
Line 20: change “data for further study” by “data for further studies”
Line 323: change “Several study” by “Several studies”
Line 28: change “affect” by “affect”
Author Response
Thank you for your letter and for the reviewers’ comments concerning our manuscript entitled “Transcriptome analysis of MyoD1 gene knockout in MDBK cell lines based on high-throughput sequencing” (ID: 1698225). Those comments are all valuable and very helpful for revising and improving our paper, as well as the important guiding significance to our researches. We have studied comments carefully and have made correction which we hope meet with approval. Revised portion are marked in red in the paper. The main corrections in the paper and the responds to the reviewer’s comments are as flowing:

Round 3
Reviewer 2 Report
The manuscript has improved but still the rationale, hypothesis and purpose of the study are not clearly established, integrating both the in vitro and in vivo approaches of the study and still the discussion and conclusions needs to be re-written, which in parts is illegible and highly speculative, for instance:
Lines 295-297: “PDE1B and ADAMTS1 was triggered after knockout MyoD1 gene, the reason may be that they were compensatory relationship in regulating muscle formation”
Lines 298-299: “is highly expressed in leg muscles, which we infer is one of the key parts of meat and this gene contributes to meat quality characteristics”
On the other hand, carefully editing for grammar and typographical errors is needed, as in spite of the authors state the corrections have been made, some are not, besides of adding more in the new paragraphs.